# Predictive System Implementation to Improve the Accuracy of Urine Self-Diagnosis with Smartphones: Application of a Confusion Matrix-Based Learning Model through RGB Semiquantitative Analysis

**DOI:** 10.3390/s22145445

**Published:** 2022-07-21

**Authors:** Seon-Chil Kim, Young-Sik Cho

**Affiliations:** 1Department of Biomedical Engineering, School of Medicine, Keimyung University, 1095 Dalgubeol-daero, Daegu 42601, Korea; chil@kmu.ac.kr; 2College of Pharmacy, Keimyung University, 1095 Dalgubeol-daero, Daegu 42601, Korea

**Keywords:** linearity, semi-quantitative analysis, RGB, urinalysis, smartphone

## Abstract

Urinalysis, an elementary chemical reaction-based method for analyzing color conversion factors, facilitates examination of pathological conditions in the human body. Recently, considerable urinalysis-centered research has been conducted on the analysis of urine dipstick colors using smartphone cameras; however, such methods have a drawback: the problem of reproducibility of accuracy through quantitative analysis. In this study, to solve this problem, the function values for each concentration of a range of analysis factors were implemented in an algorithm through urine dipstick RGB semi-quantitative color analysis to enable real-time results. Herein, pH, glucose, ketones, hemoglobin, bilirubin, protein (albumin), and nitrites were selected as analysis factors, and the accuracy levels of the existing equipment and the test application were compared and evaluated using artificial urine. In the semi-quantitative analysis, the red (R), green (G), and blue (B) characteristic values were analyzed by extracting the RGB characteristic values of the analysis factors for each concentration of artificial urine and obtaining linear function values. In addition, to improve the reproducibility of detection accuracy, the measurement value of the existing test equipment was set to an absolute value; using a machine-learning technique, the confusion matrix, we attempted to stabilize test results that vary with environment.

## 1. Introduction

Urinalysis is one of the most frequently performed tests in medical institutions because it is an inexpensive test that can detect risk signals with respect to major metabolic functions from a urine sample [1,2]. Urinalysis has three aspects: physical characteristics, such as urine color; chemical measurement; and microscopic examination [3,4,5]. The physical and chemical tests of urine are performed using a urine dipstick, and the public can easily access urinalysis. Although urinalysis is simple, it requires a significant amount of attention due to the problems, such as infection, that may occur during the collection and testing process [6]. In particular, because the test results are obtained through chemical and enzymatic reactions, errors in results may occur owing to reaction times and environment changes [7].

The dipstick used for urinalysis is a plastic strip that has pads containing reagents attached to it [8]. Therefore, by using a dipstick, pH, protein, occult blood, glucose, leukocytes, esterase, nitrites, bilirubin, and ketones can be simultaneously measured [9]. The dipstick is mainly used for rapid point-of-care testing because it has high sensitivity, a low cost, and confirms results through visual color analysis [10].

In general, color analysis with the naked eye significantly affects the reading. Brightness is important and white light of approximately 1000 lux is excellent [11]. During visual analysis, if the surrounding environment is too dark, the color will appear dark, whereas if too bright, the color will appear pale, which may cause errors in readings [12]. Therefore, an objective system that can clearly observe and evaluate dipstick color change is needed. To increase the accuracy of the test, a mechanical rather than a manual test is required. In particular, a more accurate method for detecting quantitative changes in urine chemical elements by change in dipstick color is needed. Considering the speed and accuracy of the test, the method can be applied using a camera and an application (app) in tandem in mobile devices.

This study intends to develop an algorithm and app for urinalysis RGB quantitative analysis using a smartphone to provide a solution to the aforementioned problems and improve accessibility. First, the range of items that can be inspected with a dipstick was specified and RGB values were extracted for the specified items on the basis of smartphone camera images. In addition, a method for semi-quantitatively deriving the color conversion range by comparing the extracted RGB values with standard RGB values was studied. When using the smartphone, to maintain the reproducibility of the measurement value, which varied with environmental changes, the measurement value of the existing test equipment was set as an absolute value. In addition, this study aimed to increase the reproducibility of accuracy by applying a confusion matrix-based learning model to the results of a urine test obtained using a smartphone camera [13,14].

Previous studies have focused on quantitatively converting the color conversion value of the dipstick, and there was a difference in setting the quantitative range of color similarity in this process [15,16]. Color-measurement sensitivity is based on the accuracy of the camera’s complementary metal-oxide-semiconductor sensor, the color uniformity of the color response, and the number of calibration points [17]. Herein, the method of inputting the model of the converted color through image processing and the multidimensional color conversion method through image processing were applied [18]. However, there was fluidity in the change value according to the environment. Therefore, a quantitative presentation of the radical RGB values was necessary. In addition, the proposed methods suggest specific values to maintain the objectivity of extracted values in relation to camera characteristics [19].

However, if the RGB values recognized by the smartphone camera change depending on the concentration of the object detected in the urine and the RGB is not presented quantitatively, there is a high possibility of error in the measurement value. Therefore, this study aimed to present data within a predictable range by measuring RGB change values according to the amounts of the analysis factors to be tested. In addition, the accuracy of the algorithm was ascertained by evaluating the agreement between the results of the existing test device used in hospitals and the results obtained using the app. Since urinalysis is analyzed as a chemical reaction, the impact of the surrounding environment should be minimized, and thus the test should feature a quantitative evaluation factor [20]. Therefore, an algorithm that can infer changes according to the measurement medium is required. To maintain a good agreement between the smartphone-measured value and the absolute value measured by the existing equipment, an attempt was made to increase the accuracy of reproducibility according to the test environment using a confusion matrix-based machine-learning model. This study aimed to develop a smartphone app to increase the accessibility of real-time urine tests at home and to introduce the basic mechanism of a new color analysis algorithm as a method to increase the reliability of test results.

## 2. Materials and Methods

The urine dipstick color table is presented as a color comparison table for conventional visual recognition, as shown in Figure 1 [21]. A method for comparing and tracking the same color is suggested, but distinguishing and recognizing color with the naked eye is limited. In this study, an experiment was performed to compare and track changes in detected RGB values. This was carried out by extracting the dipstick color change as RGB values of the photographed images and setting the reference values for each standard concentration for the test item. After photographing using a smartphone camera (iPhone 11 pro, 2021), RGB values were obtained, as shown in Figure 2, through image processing (photoshop CS6, 2020).

The smartphone used in this study had a screen size of 6.1 inches and a resolution of 1792 × 828 pixels at 326 ppi. It also included basic specifications, such as the Triple 12MP Ultra-Wide, Wide, and Telephoto cameras. The white balance was set automatically under the condition of lighting from a fluorescent light bulb. The suggested smartphone camera was used without changing the camera measurement sensitivity and color correction and by applying the same steps that an actual user would use. The distance from the smartphone was set to 50 ± 0.5 mm, and the average illuminance was 1000 lux (the color temperature was 4000 K). The urine dipstick used was the UroColor 10 Urine Test Strip made by Abbott Korea Diagnostics [22]. Comparative analysis of the results was performed using a urine dipstick reading device URiSCAN Pro (YD-Diagnostics, 2006, Yongin-si, Korea) used in hospitals.

The urine used in the experiment was based on artificial urine [23]. The analysis factors selected were pH, glucose, ketones, hemoglobin, bilirubin, protein (albumin), and nitrites. According to these test results, various diseases, such as carbohydrate disorders, liver and hemolysis disorders, and urinary tract infections, can be successfully suspected [24]. The urine used herein was prepared by dissolving 0.2, 8, 1.14, 0.2, and 0.05 g of KCl, NaCl, Na_2_HPO_4_, KH_2_PO_4_, and food coloring (yellow) in 1 L of triple distilled water, respectively. In addition, the pH was adjusted to 6.0 by adding HCl. Thereafter, RGB analysis was performed by adjusting the concentrations of the relevant analyte factors. For example, in the case of glucose among the analysis factors, first, a 300 mg/dL solution was prepared by dissolving 300 mg of glucose in 100 mL of artificial urine. Subsequently, 50 mL of this glucose solution was mixed with 25 mL of the artificial urine to make a diluted solution of 200 mg/dL. Subsequently, 50 mL of this prepared solution (200 mg/dL) was mixed with 50 mL of artificial urine and diluted twice to prepare 100 mg/dL. By diluting the concentration of the analysis factor, the algorithm was implemented by analyzing the characteristics and change trend of the RGB values according to the concentration change. Through this, the program was structured to extract the results in the same way as the existing results.

To reproduce the accuracy of the values measured by the smartphone camera and those measured by the existing test equipment, the measured values of the existing test equipment were set as the reference values. In addition, all specific values of RGB were set as standard values, and a learning model was implemented through the application of the confusion matrix [25].

Therefore, the measured value, which was calculated as a similar boundary value, was implemented to match the existing specific value through the learning model. RGB values implemented in actual urine tests are widely distributed; as these values are generally presented as figures that are difficult to distinguish from each other, the definition of boundary values may be ambiguous. Therefore, the measured values should be corrected to reduce errors in measurements.

To overcome this limitation, by analyzing the properties of heterogeneous RGB data values using a learning model, accuracy can be increased by reducing errors in determination and measurement values [26]. The confusion matrix comprises four sub-elements: TP (True Positive), FP (False Positive), TN (True Negative), and FN (False Negative) [27]. In this study, to obtain the same results as those of the existing test tools by setting the testing dataset and the training dataset, the decision value approximate to the testing dataset was presented as a measurement value and subsequently learned by the model. Therefore, for multiple users, the same dipstick should be tested more than 1000 times to maintain accuracy.

Accuracy was confirmed by comparing the results of the testing equipment used in existing clinical practice with those of the test conducted using the developed app. Thus, urine dipstick agreement was verified with a Bland–Altman plot [28]. The degree of agreement was confirmed based on the graphic distribution of the average value, which was obtained by performing the test 10 times under the same conditions.

## 3. Results

Changes in RGB values according to the concentration of analysis factors in artificial urine were observed. First, the change in RGB value according to pH concentration is shown in Figure 3. The RGB values analyzed as colors are displayed in different regions depending on the analysis factors. This shows the color conversion characteristics according to the changes in concentrations. Considering individual characteristics, the change areas of R, G, and B appeared to be different; hence, it was difficult to distinguish them from the overall visual image change through color conversion applied to all analysis factors. Therefore, the individual areas were calculated by estimating the change value based on the optimization factor among R, G, and B by finding the color change characteristics for the concentrations of each analysis factor.

In the case of pH among the urine dipstick analysis factors, the color conversion factor was distinctly observed. The extracted R value exhibits the highest reactivity (linear coefficient and slope) according to pH change and can be calculated on this basis. In the case of glucose, the change in G value is quantitatively distributed; hence, it can be used as an index for color conversion estimation of images taken with a smartphone camera. Table 1 presents the RGB selection criteria for the quantitative changes in each analysis factor. In particular, a linear coefficient (*r*-squared) value of 0.9 or higher is presented. Herein, the linearity range for the analysis factor corresponds to the limit of color processing obtained with a real smartphone and suggests that color readings within this range are possible.

In this experiment, because the color change values of ketones exhibit the most uniform pattern, the reliability of readings can be increased. Bilirubin presents the lowest color conversion values and the RGB values are difficult to distinguish. The same pattern can be detected in visual inspection. Therefore, it is possible to apply the change values for each analysis factor by presenting the RGB values obtained as a function, thereby reducing the error between the actual and extracted values. In addition, it is possible to use an algorithm to apply to the app.

For color analysis, a method involving hue, saturation, and value (HSV) rather than RGB is used at times [29]. In general, it is known that the HSV color space model has better detection accuracy than the RGB model in image analysis because it cannot separate the effect of light from color information. However, this study aims to utilize color space information for quantitative analysis. Therefore, it is important to analyze the linearity (coefficient of determination) and slope of RGB and HSV values according to the concentrations of urine components. As shown in Figure 4, the color space values for the urine analysis factor concentrations were extracted as in RGB extraction, and the linear coefficients and slope values were calculated using a prism (GraphPad Software, Inc., San Diego, CA, USA) [30]. As a result, the RGB value and the individual HSV value were plotted according to the concentration of bilirubin, and then a graph was created. The coefficients of determination (*r*^2^) of the R and V values of bilirubin were the highest at 0.9596 and 0.9422, respectively, and the slope was approximately thrice higher. As the reactivity of the extracted value was high according to the change in concentration, the concentration could be determined based on the change in color. Therefore, as shown in the experimental results, RGB extraction analysis tends to be similar to that of HSV.

Figure 5 shows the screen of the application in which the development algorithm is applied. It was configured to express the same quantitative result as the test result of a specific item representing the same value. Figure 6 shows the degree of agreement between the test result extracted from the device used in hospitals (URiSCAN Pro) and that obtained from the app using a Bland–Altman plot. In the graph, the black dots denote the results of testing with the existing urine test device (URiSCAN Pro), whereas the white dots denote the results extracted using the app developed in this study. The closer the two points, the more likely it is that the result will be the same. It was found that the seven results for the presented urine analytes showed the same degree of agreement. Therefore, it was predicted that there would be no difficulty in applying the algorithm developed as a function using RGB analysis values to urinalysis.

The test values measured by the developed algorithm are consistent when the test is performed in identical environments; however, the measurement environment may vary depending on the user. Therefore, a confusion matrix-based model was applied which can continuously learn the measured values regardless of the current user. The results are listed in Table 2. The measurement value for the existing equipment is an absolute value, and when a new dipstick is tested a new measurement value is learned on the basis of the value measured by the current test. As a result, an average accuracy of 98.74% can be maintained.

In this study, the standard values for artificial urine analyzed through the linear semi-quantitative algorithm were consistent with the values measured using the existing equipment. Considering this result, the proposed method will help improve the reproducibility of the accuracy of data recorded by users in actual clinical trials.

## 4. Discussion

Generally, urinalysis can test for kidney disease and other systemic diseases at a low cost. Therefore, accessing the simple test method results should be convenient. A method for presenting quantitative values by acquiring test results according to the color of urine in smartphone camera images has been recently developed, and commercialization of this method is being promoted as an app [30,31,32,33]. The urine dipstick color conversion factor is recognized as a total value; therefore, a change factor may occur depending on the illuminance or environment. In addition, because the color conversion coefficient is corrected based on an estimated specific value, there may be differences between each urinalysis app [34].

In existing research, for the most part, standard color values are inputted and matching algorithms are applied. In this case, it is difficult to present an accurate quantitative reference value because errors at the boundary of the color conversion value or corresponding to an incorrectly recognized value due to environmental conditions may occur. This problem can be solved through machine learning, which matches RGB values with spectral conversion values [35,36]. In addition, it can be solved using a method that predicts linearity and suggests semi-quantitative indicators, similar to the method employed in this study.

In general, the RGB or HSV model is used for color image analysis; however, in this study, the RGB model, which has excellent accessibility because of the availability of many applicable programs, was used, and similar results were extracted and applied from comparative experimental results. Semi-quantitative analysis has the advantage of quantitatively estimating detected amounts of analytes by extracting an optimized linear relationship according to element concentration and RGB value. The app-based urinalysis in this study obtained semi-quantitative values using RGB values and then presented a new algorithm for extracting quantitative values through AI. Therefore, it is useful to provide a standard for quantitatively predicting the degree of a suspected disease, which can be presented in the same way as the test result in a hospital, thereby increasing the reliability of the examiner. In the existing urinalysis app, color interpolation is applied for accuracy and an analysis method using linear interpolation was applied [37], but this requires that all RGB values are applied. In addition, the app has its own standard algorithm which digitally extracts the color table provided by the urine dipstick company; therefore, there may be differences depending on the app [38]. When analyzing test results with colors obtained with a smartphone camera, as in this study, linearity can be found through the formula of specific color values among the extracted R, G, and B values according to the analysis factors for each concentration. If the optimized RGB value is applied, the accuracy of the color conversion value can be improved through semi-quantitative analysis.

In addition, to maintain accuracy, an AI-based measurement service is required, and because the environment and usage method differ for each user, the test results obtained through self-diagnosis require reproducibility of accuracy. Accordingly, as an effective solution, the measured and absolute values are learned using a confusion matrix, and subsequently the similar measured values are estimated and presented. Accuracy refers to the degree to which the same result value can be maintained even in the case of repeated tests, not the classification of boundary values.

In general, color interpretation of digital images using a smartphone camera is influenced by the choice of color space and the method of measuring color similarity [39,40]. The color space provides specific values for all expressed colors, and in this study, RGB, the basic color space used in camera sensors and computer displays, was applied as a standard. The linearity of the RGB values was used to determine the similarity between colors, thereby yielding the same values as the actual test values. Therefore, the same value can be presented in a constant environment, and the accuracy of the test can be improved by applying the semi-quantitative linearity. In addition, in the case of a variable environment, the machine-learning model was applied. Nevertheless, this study has a limitation: although artificial urine was produced and quantitative values for each concentration were determined, comparison and verification of results for the urine of patients with actual diseases will be of paramount importance in future studies.

## 5. Conclusions

In this study, an algorithm was developed to extract the color of a urine dipstick from a smartphone camera image and apply the optimized RGB specific color value through semi-quantitative analysis. By extracting the RGB linear coefficient for each urine concentration of seven urine dipstick analysis factors, we succeeded in deriving result values with the same pattern as existing test results. Therefore, both the accuracy of the test result extracted based on color analysis and the error of the color analysis quantitative value in the existing algorithm were improved. In addition, considering a variable measurement environment for each user, a confusion matrix-based learning model was applied to maintain the reproducibility of accuracy; consequently, an average accuracy of 98.74% was maintained. These findings contributed to increasing the accessibility of test results and may be applied in future medical platforms.

## Figures and Tables

**Figure 1 sensors-22-05445-f001:**
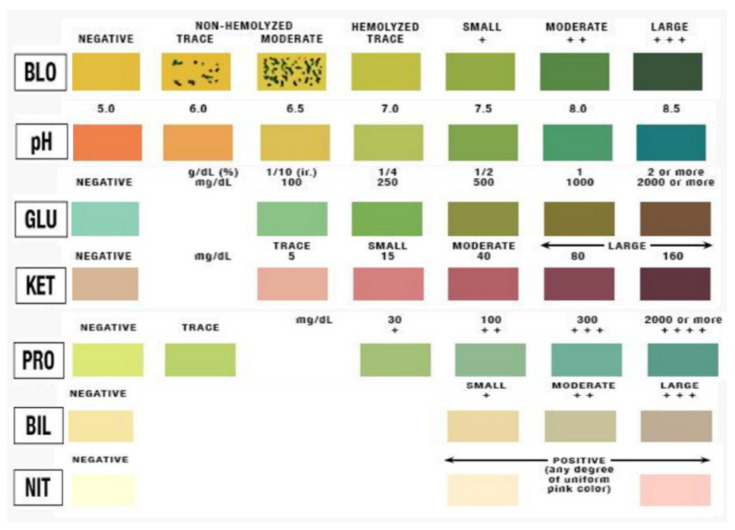
Urine dipstick color comparison test table.

**Figure 2 sensors-22-05445-f002:**
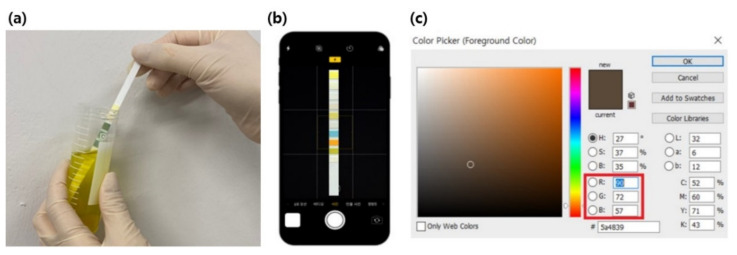
RGB extraction process for the urine dipstick. (**a**) The step where a dipstick was dipped into an artificial urine sample containing analytes. (**b**) The step where a dipstick was taken with the smartphone. (**c**) The step of extracting RGB values from the color pad of the dipstick.

**Figure 3 sensors-22-05445-f003:**
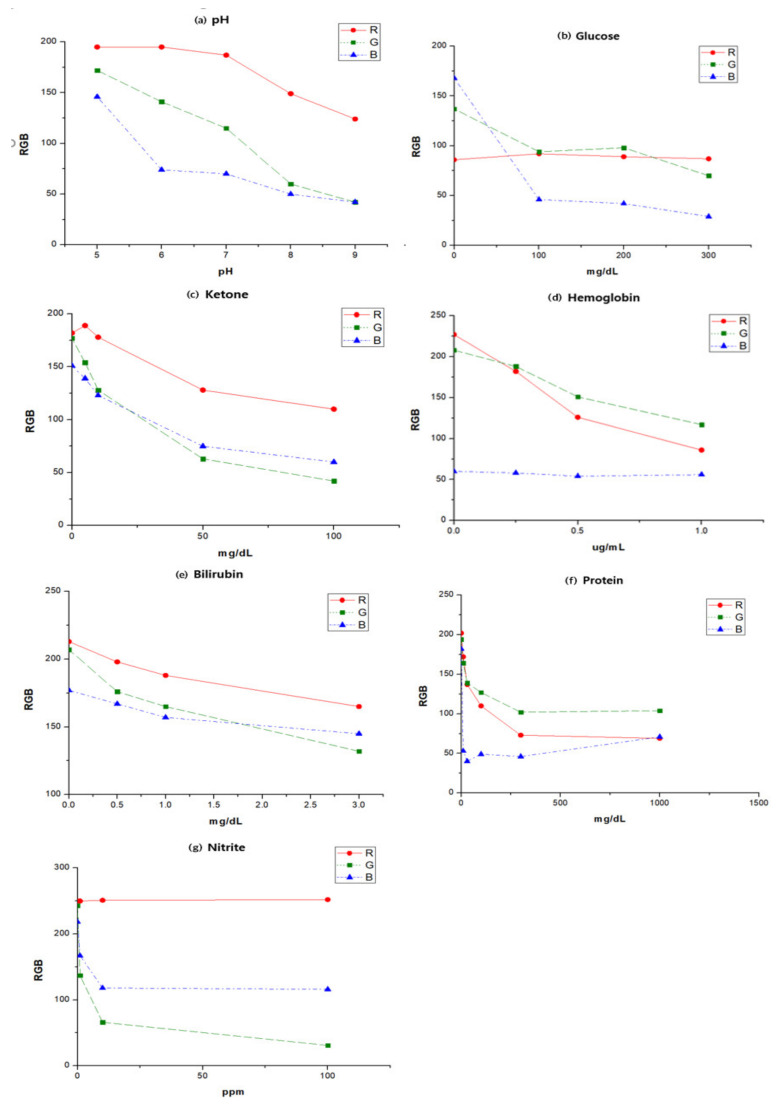
RGB change values for concentrations in the urine dipstick.

**Figure 4 sensors-22-05445-f004:**
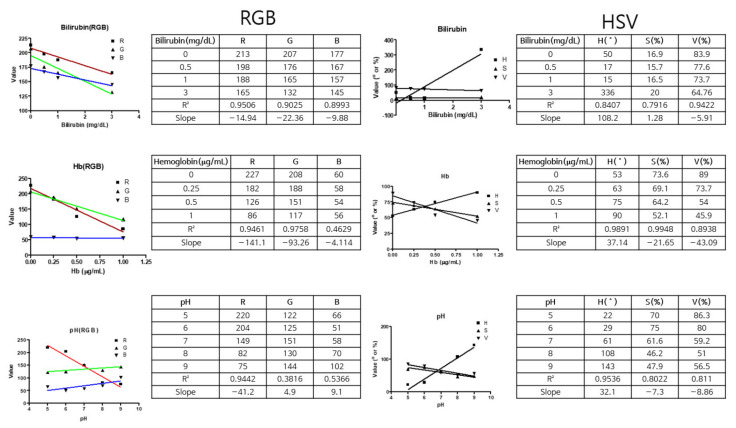
Comparison of RGB and HSV values for Bilirubin, Hb, and pH. (The values were extracted and compared using GraphPad software; an iPhone 11 pro was used as the smartphone.)

**Figure 5 sensors-22-05445-f005:**
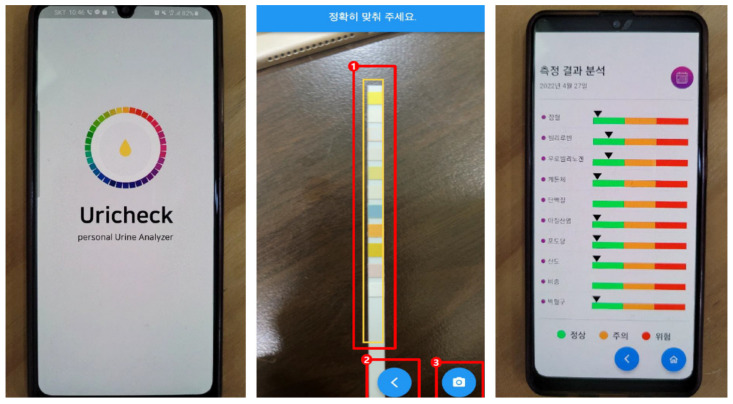
Urine dipstick test application. 1. Region of interest 2. Button for backward movement. 3. Button for taking a picture.

**Figure 6 sensors-22-05445-f006:**
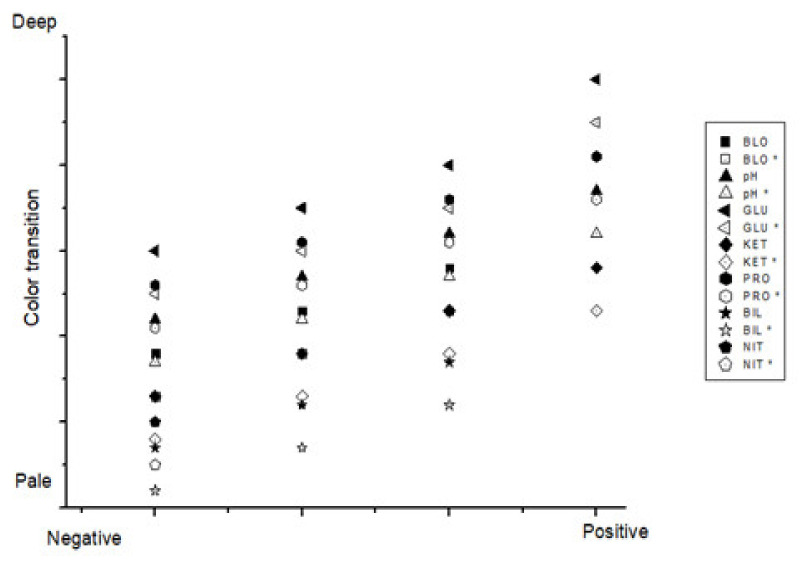
Comparison for the analysis of urine dipstick test result agreement.

**Table 1 sensors-22-05445-t001:** RGB characteristic values of analysis factors by urine dipstick concentration.

No.	Urine Analytes	R/G/B (Optimized)	Formula	Slope	R-Squared	Linearity	Reference Range
1	pH	R	Y = −40.9 × X + 432.7	−40.9	0.9442	pH 5–9	4.8–7.4
2	Glucose	B	Y = −0.3983 × X + 122.8	−0.3983	0.9104	50–300 mg/dL	<20 mg/dL
3	Ketones	G	Y = −69.92 × log X + 189.1	−69.92	0.9872	5–100 mg/dL(log)	<5 mg/dL
4	Hemoglobin	R	Y = −141.1 × X + 217	−141.1	0.9461	0.25–1.0 ug/mL	-
5	Bilirubin	G	Y = −22.36 × X + 195.2	−22.36	0.9025	0.5–3.0 mg/dL	<0.2 mg/dL
6	Protein (albumin)	R	Y = −48.88 × log X + 208.3	−48.88	0.9682	10–1000 mg/dL(log)	<2 mg/dL
7	Nitrites	G	Y = −53 × log X + 131	−53	0.963	1–100 mg/dL(log)	-

**Table 2 sensors-22-05445-t002:** Results of application of the urine dipstick confusion matrix model.

New				URiscan Pro				Smartphone				Accuracy
New 1	New 2	New 3	New 4	Test 1	Test 2	Test 3	Test 4	Train 1	Train 2	Train 3	Train 4	98.56%
New 1	New 2	New 3	New 4	Test 1	Test 2	Test 3	Test 4	Train 1	Train 2	Train 3	Train 4	97.67%
New 1	New 2	New 3	New 4	Test 1	Test 2	Test 3	Test 4	Train 1	Train 2	Train 3	Train 4	99.85%
New 1	New 2	New 3	New 4	Test 1	Test 2	Test 3	Test 4	Train 1	Train 2	Train 3	Train 4	98.88%
Average												98.74%

## Data Availability

Not applicable.

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
