# Peer review of "Predictive System Implementation to Improve the Accuracy of Urine Self-Diagnosis with Smartphones: Application of a Confusion Matrix-Based Learning Model through RGB Semiquantitative Analysis"

_sensors, 2022, doi:10.3390/s22145445_

Round 1

Reviewer 1 Report

Dear Authors,

I liked the MS ´´Predictive system implementation to improve accuracy of urine self-diagnosis with smartphone: Application of confusion matrix-based learning model through RGB semiquantitative analysis´´.

The overall idea and concept are good for the development of patient stratification in a quick time frame.   

I have some doubts about this methodology

1- each mobile cam is a slightly different setting for capturing the picture and which will lead to different pixels for RGB pictures. is there any frame of the window to define the required number of pixels for each colour to apply for stratification. What is the minimum and max limit for each RGB?

2- what is the standard error with known sample of lab-made urine?

3- What is your opinion on the intermediate place between 2 colours? how much does it impact efficiency?

Reviewer 2 Report

Dear authors

This was an interesting paper about urine analysis through a phone application; research design was clear and well-conducted. In my opinion introduction must be improved with more citations about previously published works in the same field, in example:

-DOI: 10.1089/tmj.2019.0101

DOI: 10.1016/j.euf.2020.02.002

Moreover, in discussion it must be underlined that your evidence is only preliminar (pre-clinical experiment), and a new study phase (about real urine analysis) is required.

Reviewer 3 Report

Some main comments:

The approach using RGB in color analysis is not the best possible.
The authors should consider other better suiting color models.
The figure and table caption should describe more about what is actually shown.

Here more detailed comments:

Title:
Somewhat long; could be compressed

Abstract: OK
Keywords: 'RGB', 'confusion-matrix', 'smart phone', etc  missing

Introduction:

Line 36: is this relevant in this paper, I mean infection control??

Line 38: is to my mind extremely relevant: how to make the analysis process standard?
At least tell how that was done in the described experiments.

Lines 45-46: humans differ ok, but in general human vision is excellent in color comparisons;
e.g. it takes care of white balancing that is more difficult to camera.

Line 47: the authors are right: illumination is of primary interest.
However using RGB is not a good option when illumination varies.

Line 55: Mentions 'app': is that available somewhere?
Could the authors provide a link to test it?

Lines 55-65:
To my mind there is a major problem here wrt the approach of using RGB and (smartphone)
digital camera:
The images taken by smartphones are heavily processed meaning that much attention should
be given to how to use the camera and its software for color measurements.
Actually we have tried to do that, but the results were modest at most.
That is because the camera software does a lot of tuning of colors.
The authors should pay attention to how to make sure that the camera is working
in optimal way for color analysis.

Another major concern is the use of RGB color model.
That is not good for color analysis.
There are much better color models that are ideal for color analysis
because they have been designed to be such.
One is the HSI or HSV color model  (Hue, Saturation, Intensity).
It transform RGB by a quite simple formula to such that
the color (Hue) can be compared independent of intensity and saturation.
Hue is the pure color (typically in the range 0 ... 360 degrees).
Another good point is that the intensity can be used to
control the illumination or warn that it is not good enough.
BTW, low intensity results in more noise.
BTW2, some form of checking the results is important in medical (and many other)
applications: HSI gives some good possibilities to that.

Lines 70-73:
I do not quite understand this reasoning. Perhaps due to the above points.

Line 73: 'camera characteristics':  each camera and pixel has its own characteristics,
i.e. small deviations from the ideal or nominal values.
Have the authors done any image calibrations?

Line 74:  'the change value of the RGB value' what does this actually mean?

Lines 74-86:  Seems that the authors are trying to fix the RGB problem by
using some software tools that actually are not so well documented here
(the referee has read the paper..)

Figure 1: taken from external source: give the reference in the caption.
BTW, how the reference colors were extracted, by photographing or using a pdf or...?

Lines 94-95: How the smartphone camera was actually used: what were the settings etc?

Lines 95-96: What image processing functions was used with Photoshop?
BTW, the referee though that the authors will present an app in the smartphone?!
Could you clarify, what you actually mean by the app: is it something totally external
to this work described in the manuscript??

Lines: 96-98: Yes, but how about this manuscript??? How you have applied these???

Lines 98-100: The referee is now very worried because of what he already said above
about the RGB and the camera image processing pipeline: does
the authors have any control of the camera here???

Lines 100-102: How the distance was set (obviously not by the camera control??)??
How the illumination was set/measured?
What was the light source?

Figure 2: Please explain in the figure caption what is shown in the figure(s).
( THIS APPLIES TO ALL FIGURES)
Are those your photos or taken from elsewhere (where)??

Line 109: The referee though until this point that the authors are using some
well known urine fantom.
Now it seems that the authors have mixed the solution using theirs own recipe???
Please, give reference if some 'cook book' was actually used/followed.

Lines 125-128: How to reproduce accuracy of the image processing chain???
Please, give details of how you actually measured the color of the samples.
(more detailed than just telling the you used Photoshop, but how you used it).

Lines 130-134: The referee thinks that making a good measurement the first thing
is to do the measurement well, not using AI to fix basic measurement flaws.

Lines 143-144: Yes, AI need a low of data! Here only a couple of measurements???!!!
Conclusion, do not first use AI but do the homework with basic image/color processing.
After that you might not need any AI.
BTW, AI almost always confuses what is actually happening.
=>
Not at all good for clinical safety and quality.

Lines 145-149:
The authors should convince, that the results are not due to overleaning
because of the minimal data set used...

Lines 151-153: HSI graphs should be more informative and easier to interpret.

Lines 150-160: compare to HSI (or HSV model).

Figure 3: Think this presentation for HSI model.

Figure 4: explain subimages in caption.

 Lines 164-167: Yes, typically pH is related to (red-blue) color change.
This can be done more precisely in HSI color model.

Table 1: explain X and Y in table caption (and perhaps also in text).
BTW, Slope is already in the Formula.

In all HSI model could be used for every factor.

Lines 182-183: Please, give reference to the app.

Figure 5: Is this the best possible visualization mode for this data?
(much empty space and small marks): how about line graph?

Lines 187-189: what is actually 'the algorithm developed'?
Could the authors give the details?

 Lines 194-200: The referee thinks that already the use of HSI or similar
color model would help in accuracy; no need to AI.

Table 2: Are the authors sure that the results in this table are not
due to simply to overleaning?

Lines 206-207: The referee thinks that this conclusion is not valid,
or at least more detailed description of the procedure is needed.

Lines 217-221: Most problems here can be cured by using HSI or similar color models.

Lines 238-244: The referee disagrees.

Lines 245-247: Yes, do use that information...

Line 258: Please, give more details of your procedure/algorithm.

Line 266: This must be just overlearning.

Round 2

Reviewer 3 Report

TITLE:
Predictive system implementation to improve accuracy of urine
self-diagnosis with smartphone: Application of confusion matrix-
based learning model through RGB semiquantitative analysis

Too long: make shorter:
    - authors do not give detailed implementation => omit 'implementation'
    - 'Predicted system' omit, because there is not any details about 'prediction'
    - 'improved accuracy' omit, because no comparison to other methods
    - 'through RGB semiquantitative analysis' omit this unclear phrase.

Reviewer 3) Some main comments:
The approach using RGB in color analysis is not the best possible.

Reply) We understand the concern raised by the reviewer. We understand that RGB value is not the best for the
quantitative application of color change to detect urine analytes. This approach is a simple way to predict the
unknown quantity from the correlation of specific R, G or B value to increased concentration of urine analytes. In
addition, because of its excellent accessibility, this study successfully presented and commercialized this method.

Please give reference to this (in manuscript the authors claim that the app is not available as a link; so where is it?)

The authors should consider other better suiting color models. The figure and table caption should describe more
about what is actually shown.

Reply) We understand the concern raised by the reviewer. Other color models were not suitable for App., which will be loaded into a sma
rtphone.
As suggested, we have added captions for Figures 4 and 2. These corrections
can be found on lines 112-114 and lines 198-199.

Could the authors tell why the other models were not suitable?
WBT, who has done the App and where is it documented?

Here more detailed comments:

Title: Somewhat long; could be compressed
Reply) We understand the concern raised by the reviewer. However, we believe that the title contains a colon (:),
and the part that follows the colon is considered a subtitle.

See comments above how to shorten the title.

Abstract: OK
Keywords: 'RGB', 'confusion-matrix', 'smart phone', etc missing
Reply) Based on the suggestion of the reviewer, we have revised the highlighted keywords. These corrections can be found on line 27.

Introduction:
1. Line 36: is this relevant in this paper, I mean infection control??
Reply) We understand the concern raised by the reviewer. The cases where there is a risk of infection with
pathogenic microorganisms when collecting urine with a dipstick were introduced as references. Therefore, the
content of this has literature been further described as a reference.

OK

2. Line 38: is to my mind extremely relevant: how to make the analysis process standard?
At least tell how that was done in the described experiments.
Reply) We understand the concern raised by the reviewer. The content of the query is the most basic principle of
chemical urinalysis, and this study is referring to this literature with regard to this information. When dipstick
pads used for urinalysis are wetted with urine containing various analytes, the associated chemical or enzymatic
reactions occur rapidly. This is the most basic reaction system, and the approach wherein a urine test using a
dipstick is based on the principle that the reaction occurs quickly through the medium (chemical substance). We
believe that it was ideal to refer to existing mechanisms such as references.

Yes, but how those were applied in this study, or are they not relevant at all here?

3. Lines 45-46: humans differ ok, but in general human vision is excellent in color comparisons; e.g. it takes care
of white balancing that is more difficult to camera.
Reply) We understand the concern raised by the reviewer. It is already known that the human eye can distinguish
over 5% difference in brightness. In this study, the reason for studying the effect of illuminance in the examination
environment in previous studies is that it affects the evaluation of human vision.
Reference: Color Correction Technique using an Artificial Color Board and Root-polynomial Color Correction
for Smartphone-Based Urinalysis. Mutiara, N. S.; Adhi, H. S.; 2021 17th International Conference on Quality in
Research (QIR): International Symposium on Electrical and Computer Engineering. 2021. DOI:
10.1109/QIR54354.2021.9716173

The authors confuse here color and brightness.
They are seem to be unaware of the effect of the light spectrum to color (white balance etc).
(more notes on that later)

4. Line 47: the authors are right: illumination is of primary interest.
However using RGB is not a good option when illumination varies.
Reply) We thank the reviewer for the insightful comment. As highlighted by the reviewer, the RGB color space
would be affected under different illumination. Therefore, in this study, the illuminance is presented to solve these
problems, and the research results are presented through the RGB algorithm.

The results are also affected by the illuminabt spectra (more abot that later).
THE PROBLEM IS NOT SOLVED.

5. Line 55: Mentions 'app': is that available somewhere?
Could the authors provide a link to test it?
Reply) We cannot provide a separate test link. Please refer to the results of this study for the function of the app.

Why the app is not available if it is commercial as claimed above?

6. Lines 55-65: To my mind there is a major problem here wrt the approach of using RGB and (smartphone) digital
camera: The images taken by smartphones are heavily processed meaning that much attention should be given to
how to use the camera and its software for color measurements. Actually we have tried to do that, but the results
were modest at most. That is because the camera software does a lot of tuning of colors. The authors should pay
attention to how to make sure that the camera is working in optimal way for color analysis.
Reply) We understand the concern raised by the reviewer. In this study, a smartphone camera (iPhone 11 Pro,
2021) was used, and analysis was performed using the acquired image values. The camera image itself is a
preprocessing step because the change value is tracked by setting the image processing factor of the device to a
fixed value rather than a variable value. This algorithm is in the process of an international patent, and the same
is presented as a result of this study, wherein we state that there is no significant effect even if it is received rather
than not affected. Please refer to the data compared and analyzed with the reference value (hospital inspection
device).

The authors seem not to understand how a smartphone of processing the image:
it is doing a lot already under the default settings.
The authors should give the setting values and other details.
And remember that those details vary between smartphone brands and types,
even between each smartphone invividual => need calibration...
EVEN EACH PIXEL OF A GIVEN SMARTPHONE DIFFERS
=> calibration pixel wise

7. Another major concern is the use of RGB color model.
That is not good for color analysis.
There are much better color models that are ideal for color analysis
because they have been designed to be such.
One is the HSI or HSV color model (Hue, Saturation, Intensity).
It transform RGB by a quite simple formula to such that
the color (Hue) can be compared independent of intensity and saturation.
Hue is the pure color (typically in the range 0 ... 360 degrees).
Another good point is that the intensity can be used to
control the illumination or warn that it is not good enough.
BTW, low intensity results in more noise.
BTW2, some form of checking the results is important in medical (and many other)
applications: HSI gives some good possibilities to that.
Reply) We thank the reviewer for the insightful comment. The experiment was performed as shown in the table
below referring to your comments but there is no significant difference. Further the RGB model presented in this
study is considered to have sufficient research results. Please refer to the comparison table.

Thank you for the comparison: to my minf the HSV gives better results and offer
also a way to check the overall illumination.
So, could the authors explain explicitly why they think that the RGB is better.

8. Lines 70-73: I do not quite understand this reasoning. Perhaps due to the above points.
Reply) We understand the concern raised by the reviewer. This sentence refers to the literature, and as stated
previously, this sentence highlights the need for quantitative values to minimize environmental factors.

Yes, tha most important 'environmental factor' is the illumination.
That is allways so when using images to measurements like in this paper.
=>
The authors should better control the illumination.
BTW, why not use smartphones own light?
It is much better than any lamp that just happens to hang around.

9. Line 73: 'camera characteristics': each camera and pixel has its own characteristics, i.e. small deviations from
the ideal or nominal values.
Have the authors done any image calibrations?
Reply) We thank the reviewer for the insightful comment. Performing image correction is not ideal for
quantitative values. Therefore, as presented in this study, we aimed the maximize the benefits of the APP by
applying the algorithm analysis value to the acquired value, and implemented an algorithm to infer a quantitative
value through machine learning with the formula of the change estimate instead of the correction value.

The authors have totally misunderstood basics of image calibration.
Please, get familiar with image calibration.

10. Line 74: 'the change value of the RGB value' what does this actually mean?
Reply) The changes in RGB values are caused by the chemical reaction between the analyte and its corresponding
compounds on dipstick pads. Accordingly, evaluating the specific value among the RGB values that is closely
correlated to the analyte’s concentration is significant.

The authors obviously have not understood the comment:
Please give the exact formula that you have used here.

11. Lines 74-86: Seems that the authors are trying to fix the RGB problem by
using some software tools that actually are not so well documented here
(the referee has read the paper..)
Reply) The software tools used in the paper are certainly not documented. We request the reviewer to refer to the
References for the same.

Please make the documentation. The reader does not know what you have exactly done.

12. Figure 1: taken from external source: give the reference in the caption.
BTW, how the reference colors were extracted, by photographing or using a pdf or...?
Reply) The image in Figure 1. is taken from the manufacturer’s photograph. We have not performed any
extraction from the reference color. This figure indicates that reference colors for analytes at different
concentrations are simply qualitative according to this chart.

So, simply give the reference to the image source.
And also tell where you have got the colors.

13. Lines 94-95: How the smartphone camera was actually used: what were the settings etc?
Reply) Any parameters in camera are not modified for getting images.

Give the default setting, please.

14. Lines 95-96: What image processing functions was used with Photoshop?
BTW, the referee though that the authors will present an app in the smartphone?!
Could you clarify, what you actually mean by the app: is it something totally external
to this work described in the manuscript??
Reply) We understand the concern raised by the reviewer. In Photoshop, only the RGB values were extracted.

So, how did you do taht exactly?
By what function. Did you use any ROI, etc?

15. Lines: 96-98: Yes, but how about this manuscript??? How you have applied these???
Reply) The reference was only cited to state how other studies are solving the degree of color response.

How about telling that earlier in the manuscript?

16. Lines 98-100: The referee is now very worried because of what he already said above about the RGB and the
camera image processing pipeline: does the authors have any control of the camera here???
Reply) We understand the concern raised by the reviewer. The images are taken using the smartphone camera
under default settings.

Please tell, what the default setting were.

17. Lines 100-102: How the distance was set (obviously not by the camera control??)?? Reply) distance from
dipstick on the white background board.
How the illumination was set/measured? Reply) lighting instrument (?).
What was the light source? Reply) fluorescent light bulb on the lab ceiling.

OK, THIS IS VERY IMPORTANT INFORMATION:
THERE SEEMS TO BE NOT ANY DISCIPLINED CONTROL OF ILLUMINATION.
USING FLUORESCENT BULB in COLOR MEASUREMENT IS NOT GOOD:
WHY?
BECAUSE OF THE FB SPECTRA THAT DISTORTS COLOR!!!

18. Figure 2: Please explain in the figure caption what is shown in the figure(s).
(THIS APPLIES TO ALL FIGURES)
Are those your photos or taken from elsewhere (where)??
Reply) This figure demonstrated the process of RGB extraction from the dipstick that is dipped into artificial
urine sample. The image at the left shows the step that dipstick is dipped into artificial urine samples containing
analytes. Next, the image of colored dipstick inside a square box is taken using smartphone (middle picture).
Finally, the corresponding RGB number is extracted from several points of colored pad on the dipstick (right
picture). This is a picture we all took ourselves in this study. We have added the caption below Figure 2. These
corrections can be found on lines 112-114.

OK, the authors have used iPhone, not any other smartphone?
How about that in the manuscript title?
The authors claim 'smartphone' but actually mein iPhone, only?

19. Line 109: The referee though until this point that the authors are using some
well known urine fantom.
Now it seems that the authors have mixed the solution using theirs own recipe???
Please, give reference if some 'cook book' was actually used/followed.
Reply) The artificial urine was prepared according to the recipe as described previously [in text citation]. This
document has been added as reference number 23 in the manuscript.
Refer: Khan, L.B.; Read, H.M.; Titchie, S.R.; Proft, T. Artificial Urine for Teaching Urinalysis Concepts and
Diagnosis of Urinary Tract Infection in the Medical Microbiology Laboratory. J. Microbiol. Biol. Educ. 2017, 18,
2, pp. 1-6. DOI: https://doi.org/10.1128/jmbe.v18i2.132

Excellent!

20. Lines 125-128: How to reproduce accuracy of the image processing chain???
Please, give details of how you actually measured the color of the samples.
(more detailed than just telling the you used Photoshop, but how you used it).
Reply) We understand the concern raised by the reviewer. In the Materials and Methods section, a series of
dipstick dipping, image capture, and RGB extraction are described in detail. Theoretically, the changes in color
of each pad is caused from chemical reaction between reactive chemicals and urine analytes. Therefore, the
chemical reaction and its associated color changes are proportionate to the analyte’s concentrations. The color
changes according to the function of analytes is expected to be reproducible unless unknown interference
ingredients are included in urine.

The authors have missed the point:
the referee means the image processing chain in the software.
Please give details of that.

21. Lines 130-134: The referee thinks that making a good measurement the first thing is to do the measurement
well, not using AI to fix basic measurement flaws.
Reply) The aim of this study is to find and reduce the causes of fluctuations in the measured values, and thus, this
method is presented.

Yes, but for that you should do more experiments using different
smartphones, illuminations, etc.

Now only a couple of test with one camera and one light???

=>  do more tests to show that the approach is reducing fluctuations,

22. Lines 143-144: Yes, AI need a low of data! Here only a couple of measurements???!!! Conclusion, do not first
use AI but do the homework with basic image/color processing. After that you might not need any AI.
BTW, AI almost always confuses what is actually happening.
=> Not at all good for clinical safety and quality.
Reply) Although we appreciate the comment of the reviewer, unfortunately, we do not entirely agree with the
same We are not entirely sure what the reviewer is implying when he/she is referring to “basic color processing.”
The AI will be improved based on the learning effect, and the results of this study suggest the reproducibility of
accuracy. In addition, the authors conducted basic research for a long time, and suggested an improved method
based on this research.

'Basic color processing' in a way we do agree: however, perhaps not in the way the authors think.
=> please do your homework with basic (elementary) color processing methods.

23. Lines 145-149:
The authors should convince, that the results are not due to overleaning
because of the minimal data set used...
Reply) We believe that a higher volume of data would be acquired under varying conditions such as illumination,
background, and smartphone device.

Please show explicitly, using larger data set that you approach really works.
Now you claim that is works while it most likely is doing only overlearning.

24. Lines 151-153: HSI graphs should be more informative and easier to interpret.
Reply) When RGB was converted to the HSV value for quantitative analyses, RGB data seemed to be more
reliable than HSV in terms of slope and r2 (coefficient of determination).

How is that?

25. Lines 150-160: compare to HSI (or HSV model).
Reply) As mentioned previously (question 24), a change in either of R, G, or B value is more responsive to
different concentrations of each ingredient in urine. Please refer to reply 7.

How is that?

26. Figure 3: Think this presentation for HSI model.
Reply) When some data applied in HSV model, RGB model is superior or equivalent to HSV.

How is that?

27. Figure 4: explain subimages in caption.
Reply) We understand the concern raised by the reviewer. The following is the representation for the subimages:
â‘  region of interest â‘¡ button for backward moving â‘¢ button for taking a picture. We added these
descriptions to the captions. These corrections can be found on line 198-199.

OK

28. Lines 164-167: Yes, typically pH is related to (red-blue) color change. This can be done more precisely in HSI
color model.
Reply) We request the reviewer to refer to responses to concerns 7, 24, and 25.

Disagree...

29. Table 1: explain X and Y in table caption (and perhaps also in text).
BTW, Slope is already in the Formula.
Reply) The X and Y indicate analyte’s concentration and extracted RGB number, respectively.

OK

30. In all HSI model could be used for every factor.
Reply) The RGB number can be easily transformed into an HSV number. Under ambient illumination, specific R,
G, or B number is correlated with increased concentrations of analytes, such as pH, hemoglobin, and bilirubin.

Yes, so do that...

31. Lines 182-183: Please, give reference to the app.
Reply) The figure represents the app screen, and it is similar to a sentence.

???

32. Figure 5: Is this the best possible visualization mode for this data?
(much empty space and small marks): how about line graph?
Reply) This graph shows the change in RGB values.

So, what is your answer: is this really the best possible way???

33. Lines 187-189: what is actually 'the algorithm developed'?
Could the authors give the details?
Reply) The authors believe that by referring to the results, the developed algorithm can be easily understood. In
addition, the mathematical solution of the algorithm is presented as a formula.

Please give it explicitly.
(promised so already in title)

34. Lines 194-200: The referee thinks that already the use of HSI or similar
color model would help in accuracy; no need to AI.
Reply) Although we appreciate the insightful comment, we do not entirely agree with the same. In the initial
stages of this study, we identified the problems of existing products with the change value of environmental factors.
This information is presented in the introduction section.

Yes, but you have not shown that in this paper:
make more test to convince the readers also.

35. Table 2: Are the authors sure that the results in this table are not
due to simply to overleaning?
Reply) The authors are certainly confident about the results of this study.

Yes, but give also clear evidence that the results are not overleaned ones.

36. Lines 206-207: The referee thinks that this conclusion is not valid,
or at least more detailed description of the procedure is needed.
Reply) We understand the concern raised by the reviewer. We hope that the reviewer understands the information
is presented based on the method and results of this study.

Disagree

37. Lines 217-221: Most problems here can be cured by using HSI or similar color models.
Reply) A few comments related to HSI (V) were previously answered (questions 24, 25, and 26).

Disagree

38. Lines 238-244: The referee disagrees.
Reply) From an accurate indication of the errors of the proposed research method, we agree with your opinion.
However, we do not agree that we have presented the results of this study without logic.

This is not an exercise of logic but measurements:
please, give error estimates.

39. Lines 245-247: Yes, do use that information...
Reply) As stated in a previous response, no prior image processing of the camera was performed, and it was
configured using the default settings.

The camera itself is doing many image processing operations before releasing the image.
This may affect the results and vary between camera types...

40. Line 258: Please, give more details of your procedure/algorithm.
Reply) We have presented this information in the Methods section of this study.

No, I do not think so.

41. Line 266: This must be just overlearning.
Reply) We request the reviewer to refer to the study methods and results.

That does not tell anything about overlearning:
The author sshould do their home work with overleaning
and how to avoid it.
